# Effects of Sulfamethoxazole and 2-Ethylhexyl-4-Methoxycinnamate on the Dissimilatory Nitrate Reduction Processes and N_2_O Release in Sediments in the Yarlung Zangbo River

**DOI:** 10.3390/ijerph17061822

**Published:** 2020-03-11

**Authors:** Huiping Xu, Guanghua Lu, Chenwang Xue

**Affiliations:** 1Key Laboratory of Integrated Regulation and Resource Development on Shallow Lakes of Ministry of Education, College of Environment, Hohai University, Nanjing 210098, China; huiping__xu@163.com; 2Water Conservancy Project & Civil Engineering College, Tibet Agriculture & Animal Husbandry University, Linzhi 860000, China; xuechenwang1@163.com

**Keywords:** high-altitude river, PPCPs, denitrification, anammox, N_2_O release

## Abstract

The nitrogen pollution of rivers as a global environmental problem has received great attentions in recent years. The occurrence of emerging pollutants in high-altitude rivers will inevitably affect the dissimilatory nitrate reduction processes. In this study, sediment slurry experiments combined with ^15^N tracer techniques were conducted to investigate the influence of pharmaceutical and personal care products (alone and in combination) on denitrification and the anaerobic ammonium oxidation (anammox) process and the resulting N_2_O release in the sediments of the Yarlung Zangbo River. The results showed that the denitrification rates were inhibited by sulfamethoxazole (SMX) treatments (1–100 μg L^−1^) and the anammox rates decreased as the SMX concentrations increased, which may be due to the inhibitory effect of this antibiotic on nitrate reducing microbes. 2-Ethylhexyl-4-methoxycinnamate (EHMC) impacted nitrogen transformation mainly though the inhibition of the anammox processes. SMX and EHMC showed a superposition effect on the denitrification processes. The expression levels of the denitrifying functional genes *nirS* and *nosZ* were decreased and N_2_O release was stimulated due to the presence of SMX and/or EHMC in the sediments. To the best of our knowledge, this study is the first to report the effects of EHMC and its mixtures on the dissimilatory nitrate reduction processes and N_2_O releases in river sediments. Our results indicated that the widespread occurrence of emerging pollutants in high-altitude rivers may disturb the nitrogen transformation processes and increase the pressure of global warming.

## 1. Introduction

To satisfy the growing industrial and agricultural demand of human beings, much anthropogenic nitrogen is lost to the air, water and land, and this has drastically disrupted the global nitrogen cycle [1,2]. Excessive nitrogen load is an important driver of water pollution (e.g., harmful algae blooms, eutrophication, and seasonal hypoxia) [3]. The dominant nitrogen removal pathways in aquatic systems are the denitrification and anammox processes [4,5]. Denitrification and anammox permanently removal nitrogen from aquatic ecosystems by converting nitrates into dinitrogen (N_2_) [6]. However, N_2_O, as an intermediate product, is produced during the denitrification processes, and the N_2_O in the atmosphere is, at present, 19% higher than pre-industrial levels [7]. In addition, N_2_O is difficult to remove in the atmosphere, given its steady-state life span of approximately 120 years. The release of N_2_O is recognized as a cause of climate warming [8,9]. 

Many scholars have studied nitrogen pollution in typical rivers, lakes and the marine environment [10,11,12,13], but information on the nitrogen transformation processes in high-altitude aquatic systems is still lacking. The Yarlung Zangbo River, the highest and longest plateau river in China, is under the influence of human activities [14,15]. The nitrogen removal processes in high-altitude areas are a nonnegligible part of the global nitrogen sink [16]. Therefore, the nutrient nitrogen pollution of the Yarlung Zangbo River has attracted extensive attention.

Pharmaceutical and personal care products (PPCPs), as a kind of emerging pollutant, are ubiquitous in freshwater ecosystems worldwide and can disrupt ecological processes and functions [17]. Recently, many PPCPs were detected in high-altitude aquatic environments, of which the antibiotics and organic ultraviolet filters (OUVFs) are two typical PPCPs [18]. The antibacterial properties of antibiotics may affect microbial activity in aquatic environments. Previous studies have reported that residual antibiotics in estuarine and coastal sediments can influence nitrate reduction processes owing to their influence on the activity and community composition of the denitrifying bacteria [19,20]. However, there has been little research on the effects of antibiotics on the nitrogen transformation in high-altitude river sediments. OUVFs are a vital additive in personal care products (e.g., sunscreen and cosmetics) and are often continuously released into the environment. Some recent studies showed that OUVFs could influence the enzyme activities that are responsible for the transmission of nerve impulses and that counteract damage caused by reactive oxygen species in organisms. Additionally, OUVFs can compete with verapamil (VER, a model inhibitor of P-glycoprotein) for binding site of *Tetrahymena thermophila* P-glycoprotein, and thereby inhibit the multixenobiotic resistance activity in *Tetrahymena thermophila* [21]. However, the influence of OUVFs on nitrogen transformation and denitrifying bacteria has not been reported. 

Sulfamethoxazole (SMX) was a typical sulfonamide antibiotic. It was detected frequently in surface water, sewage and soil [22,23,24]. The sulphonamides have a low affinity for solid and a small solid-liquid distribution (K_d_). Due to the low K_d_, sulphonamides has highly mobile in sediments which causes a great risk of water pollution [25]. 2-Ethylhexyl-4-methoxycinnamate (EHMC) was a frequently used ultraviolet filter in the sunscreens and other cosmetic products. EHMC is not only widely distributed in aquatic environments but also accumulates in aquatic organisms [26,27]. It has been reported that EHMC and its degradation products shown high ecological toxicity [28]. According to our previous studies, it is found that antibiotics and OUVFs occurred and were widely occurrence and distributed in the Yarlung Zangbo River. Meanwhile, SMX and EHMC are the main components among the antibiotics and OUVFs. The coexistence of multiple pollutants and their interactions make the pollution situation more complicated in aquatic ecosystems more complicated. Undoubtedly, the focus on the impacts of combined pollutants on nitrogen transformations in aquatic ecosystems need to be highlighted, especially in the relatively fragile plateau rivers. In this study, sulfamethoxazole (SMX), a typical sulphonamide antibiotic, and 2-ethylhexyl-4-methoxycinnamate (EHMC), a frequently used ultraviolet filter in sunscreens and other cosmetic products, were selected as representative PPCPs to explore their combined effects on the nitrogen transformation processes and the rate of N_2_O release, as well as the abundances of the denitrifying related genes (*nirS* and *nosZ*) in the Yarlung Zangbo River sediments. This study provides comprehensive information on the effects of PPCPs on nitrogen conversion in high-altitude rivers and provides a basis for assessing the ecological and environmental risks associated with PPCPs in aquatic ecosystems.

## 2. Materials and Methods

### 2.1. Chemicals and Reagents

The CAS number of sulfamethoxazole (SMX) is 723-46-6. The CAS number of 2-ethylhexyl-4-methoxycinnamate (EHMC) is 5466-77-3. SMX and EHMC were supplied by ANPEL Laboratory Technologies Inc. (Shanghai, China), and their purity levels were 98%. HPLC-grade organic solvents (acetonitrile, methanol and acetone) and analytical-grade drugs (potassium chloride, aluminium chloride and formic acid) were supplied by Nanjing SenBeiJia Biological Technology Co., Ltd. (Nanjing, China). 

### 2.2. Sample Collection and Pretreatment

The near-bottom water and the surface sediments were collected from the Linzhi Section of the Yarlung Zangbo River (latitude of 29.44° N and longitude of 94.54° E) in May 2019, at an altitude of 2910 m. May corresponds to the high flow season of the Yarlung Zangbo River in which there are abundant source of nitrogen. The water quality parameters and sediments physicochemical properties at the sampling point are presented in Table 1. The organic glass hydrophore were used to hold forty litres of water. The surface sediment samples (0–5 cm) were collected with a mud miner. All the samples were stored around the ice and returned to the laboratory within 2 h. The concentrations of SMX and EHMC in the sediments were measured. The ambient SMX and EHMC were removed by a pre-incubation experiment that lasted approximately one month [29]. Briefly, the collected sediment samples were incubated in open glass containers, and the in-situ water flow was maintained by a multi-channel peristaltic pump. After pre-incubation, SMX and EHMC could not be detected in the sediments. Subsequently, the sediments were homogenized under an argon atmosphere for the subsequent experiments. 

### 2.3. Slurry Incubation Experiments for the Removal of SMX and EHMC

To understand the removal dynamics of SMX and EHMC during the slurry incubation, pollutant-added slurry incubation experiments were performed. SMX and EHMC were added to methanol to prepare the stock solutions, and the stock solutions were added to the deionized water to make individual solutions with concentration of 10 μg L^−1^. The optimum concentration was determined by a preliminary experiment. These solutions were purged with argon for 15 min to remove dissolved gases from the solutions. The contaminant solutions were served in the gastight borosilicate vials (120 mL) and the vials were sealed to prevent leakage of the solutions and gases [30]. These vials serve as the control group. The contaminant solution were mixed with the pre-incubated sediments at a sediment-to-water ratio of 1:7 in confined spaces that were filled with argon, which was similar to the slurry incubation experiments conditions [31]. And then, the slurries were stirred vigorously to homogenize them. The slurries were divided into two groups. To one group, 200 μL of a saturated mercury chloride solution (HgCl_2_) was added to inhibit the activity of the microorganisms [32]. This group was designated the abiotic group. The aim of this group was to reveal the adsorption of SMX and EHMC by the sediments. Another group, containing only the filled slurries, was set up as the biotic group to measure the biotic degradation of SMX and EHMC. All of the treatments had three replicates and the culture temperature was 15 °C in the dark. At incubation times of 0, 2, 4, 8, 12, 24, 36, and 48 h, the vials were collected from each group and centrifuged at 2000× *g* (10 min) to obtain the supernatants. One millilitre of the supernatant from each set of sample was withdrawn, then filtered with 0.45 μm glass fibre filters. In addition, the concentrations of the SMX and EHMC were detected using high-performance liquid chromatography-tandem mass spectrometry (HPLC-MS) in accordance with our previous work [33]. These analyses had detection limits of 0.2 and 0.3 ng L^−1^ for SMX and EHMC, respectively. The mean recovery values of SMX were 107% and the EHMC was 84%.

In the control group, the change of SMX (or EHMC) concentration was generated by the hydrolysis of SMX (or EHMC). In the abiotic group, it is assumed that all microbial activities are inhibited by HgCl_2_. Then, the concentration change in the abiotic group is determined by hydrolysis and the adsorption of sediment. In biotic group, the removal rate of SMX (or EHMC) was determined by the change in concentration before and after incubation. The difference of the concentration between the biological group and the abiotic group was caused by biodegradation. The removal rate and adsorption rate of SMX (or EHMC) are calculated using following formulas.
R_i_ = C_final_ − C_initial_,
(1)
R_h_ = R_control_,
(2)
R_a_ = R_abiotic_ − R_control_,
(3)
R_b_ = R_biotic_ − R_abiotic_,
(4)
R_r_ = R_biotic_,
(5)
where R_i_ is the concentration change rate of SMX (or EHMC) in the three groups. C_initial_ is the initial contaminant concentration in each group (10 μg L^−1^). C_final_ is the final contaminant concentration in each group after incubation. R_h_ is the hydrolysis rate. R_a_ is the adsorption rate. R_b_ is the biodegradation rate. R_r_ is the removal rate.

### 2.4. Slurry Incubation Experiments for the Nitrate Reduction Processes

The time series experiments were performed first to determine the optimal response time of the effects of SMX and EHMC to the nitrate reduction processes. Deionized water was purged with argon for 15 min. The pre-incubated sediments were mixed with deionized water (water to sediments 1:7) and then stirred to make sure slurries thoroughly mixed. The slurries were pre-cultured for 24 h to eliminate background nitrate. The experiment was conducted in the gas-tight borosilicate vials (12 mL). The ^15^NO_3_^−^ (99.2%, ^15^N) were injected into the slurry vials so that the concentration of ^15^N was 100 μmol L^−1^ in each vial. SMX and EHMC (alone or in a combination) were added to the vials so that the final concentration was 1 μg L^−1^. This concentration was chosen because it best reflects the nitrogen conversion characteristics in the sediments. The control group received slurries without pollutants. The slurries were poured into the vials, and the vials were sealed. The vials were shaken and incubated (15 °C) in the dark. At 0, 2, 4, 6, 8, 12, 24, 36 and 48 h of culture, three replicate samples were collected. The nitrogen gases (^29^N_2_ and ^30^N_2_) were measured using a membrane inlet mass spectrometer (MIMS, Bay Instruments, Easton, MD, USA).

According to the results of the above time series experiments, the test period of the nitrate reduction processes was determined to be 12 h. Five gradient concentrations of SMX and EHMC, namely, 0.01, 0.1, 1, 10 and 100 μg L^−1^, were selected to study their influences on the nitrate reduction processes. The ^15^N isotope pairing technique was used to determine the denitrification and anammox rates [34]. The slurry vials were treated the same as the time series experiments. After incubation, the saturated HgCl_2_ solution (200 μL) was injected into the vials to inhibit microbiological activity. The ^29^N_2_ and ^30^N_2_ were measured. The respective rates of assimilatory nitrate reduction (denitrification and anammox) were calculated using the following equations [19,30]:(6)$${D\;=\;2N}_{30}\;+\;\frac{N_{30\;}\times \;2\;\times \;\left({1\;-\;F}_{n}\right)}{F_{n}},$$
(7)$${A\;=\;N}_{29}\;-\;\frac{N_{30\;}\times \;2\;\times \;\left({1-F}_{n}\right)}{F_{n}},$$
where D and A (μmol ^15^N g^−1^ h^−1^) are the denitrification and anammox rates derived from the ^15^NO_3_^−^ during the incubation, respectively. N_30_ and N_29_ (μmol ^15^N g^−1^ h^−1^) indicate the total ^29^N_2_ and ^30^N_2_ production during the incubation. In addition, F_n_ is the fraction of ^15^N in the total nitrate pool.

### 2.5. Slurry Incubation Experiments of N_2_O Release

The concentration variations of the dissolved N_2_O in incubation processes were determined by the headspace equilibrium technique. The slurries were handled in the same manner as the above samples, except 25 mL vials were used. The ^15^NO_3_^−^ was injected into the vials and the vials were split evenly into two groups. One of which was the initial group which was added to the saturated HgCl_2_ solution (200 μL). Another vials without HgCl_2_ served as the termination group. Both sets of vials were incubated (12 h) under the temperature 15 °C without light. Next, the reactions of the termination group were injected the saturated HgCl_2_ solution (200 μL) in order to terminate reaction. The supernatants (5 mL) were obtained by the centrifugation from the slurries, and the supernatants were transferred to evacuated serum vials. The dissolved N_2_O concentration was determined by gas chromatography [35]. The release rates of N_2_O were calculated using equation [19]:(8)$$R\;=\;\frac{{(N}_{t}\;-{\;N}_{i})\;\times \;V}{W\;\times \;T},$$
where R (nmol N g^−^^1^ h^−^^1^) is the N_2_O release rate, N_t_ (nmol mL^−^^1^) is the total amounts of N_2_O dissolved in the termination slurry samples, and N_i_ (nmol mL^−^^1^) is the total amounts of N_2_O dissolved in the initial group, V (mL) is the volume of the vial, W (g) is the dry weight of the sediment, and T (h) represents the incubation time.

### 2.6. Gene Expression Assay

Nitrate reduction related genes encoding nitrite reductase (*nirS*) and nitrous oxide reductase (*nosZ*) were detected in the slurry to explore the molecular mechanism under the influence of SMX and EHMC. In short, the incubated slurries, which were freeze-dried, were used to extract the total DNA by a Fast DNA Spin Kit for Soil (MP Biomedicals, Cleveland, OH, USA). The *nirS* gene was amplified from the extracted DNA with primers cd3aF and R3cd [36]. Primers nosZ2F and nosZ2R were used to amplify the *nosZ* gene [18]. The expression of the two genes was determined by real-time quantitative polymerase chain reaction (qRT-PCR) system, and the detailed information is shown in Table 2. The ABI 7500 Sequence Detection System (Applied Biosystems, Carlsbad, CA, USA) was used to determine the gene copy numbers of *nirS* and *nosZ*. The standard curves for the two genes were established with a 10-fold dilution sequence of standard plasmid DNA (10^2^–10^9^ copies). All qRT-PCRs were repeated three times. The amplification efficiencies were 97.7% (R^2^ = 0.9997) for *nirS* and 98.3% (R^2^ = 0.9993) for *nosZ*. 

### 2.7. Statistical Analysis

The statistical analyses used the Statistical Package for the Social Sciences (SPSS, version-20.0, Chicaco, IL, USA). One-way analysis of variance (ANOVA) and Tukey’s test were used to evaluate the differences among treatments. Correlations between nitrate reduction rates (denitrification and anammox), the abundance of genes (*nirS* and *nosZ*) and N_2_O release were analyzed using Pearson correlation analysis. The statistically significant were considered when values of *p* < 0.05. Regression analyses were also used.

## 3. Results and Discussion

### 3.1. Adsorption and Degradation of SMZ and EHMC in the Slurries

The changes in EHMC and SMX concentration in the degradation experiments are shown in Figure 1. The EHMC concentration did not change significantly in the control group during the incubation process. The EHMC concentration in the abiotic group decreased with the incubation time, but the maximum reduction was only 6% compared with the control group, which should be attributed to the adsorption by the sediments. During the whole incubation period, the concentration of EHMC in the biotic group was always lower than those in the abiotic group, indicating that the microbial degradation has taken place. However, the degradation potential of EHMC by the microorganisms in the sediments in the Yarlung Zangbo River is relatively weak, and the removal rate by biodegradation was only approximately 4% in the slurry system within 48 h. Previous studies have also indicated that photodegradation is the main way of transforming EHMC in aquatic environments [29]. But the adsorption of sediments and biodegradation cannot be ignored in the fate of EHMC in aquatic system.

It was observed that the SMX concentration decreased during the incubation period in all the treatments (Figure 1). The decrease in SMX in the control group (8%) might be a result of hydrolysis [38]. Compared with the control group, the significant decrease in SMX concentrations (18%) in the abiotic group indicated that SMX was apparently adsorbed by the sediments. Studies have shown that the adsorption rate of sulphonamides antibiotics in typical rivers was 2.9% [18]. The reason why SMX adsorption in Yarlung Zangbo River sediments was higher than that found in typical rivers may be due to their higher pH values [39]. In the biotic group, SMX concentrations were continually reduced with incubation time, and the removal rates of SMX were always higher than those found in the abiotic group with maximum increase of 40% at 48 h. SMX reduced primarily caused by the microbial degradation and/or transformation (14%). The carbon and nitrogen sources were consumed during biodegradation of SMX and EHMC, which may compete with nitrate reduction for nutrients [40]. This indicates that the microorganisms play a robust role in the removal of SMX from the aquatic environment. Compared with EHMC, SMX was more likely to be adsorbed and retained by the sediments, but neither is readily biodegraded based on their removal dynamics. These PPCPs exist in the water and sediments may affect the nitrogen conversion in the high-altitude rivers.

### 3.2. Influence of SMX and EHMC on the Nitrate Reduction Processes

The changes in the denitrification and anammox rates with time under different treatments are shown in Figure 2. Compared to the control, the presence of EHMC and SMX significantly decreased the denitrification and anammox rates. EHMC and SMX alone exhibited similar inhibition on the denitrification and anammox rates, while the inhibitory effect produced by their mixture is more significant. Regarding the time response, the denitrification and anammox rates peaked at 12 h and then declined rapidly for all the treatments and did not significantly differ from each other after 24 h. This may be because the nutrients are consumed or the readily available electron donors are depleted in the vials [19]. Therefore, the period of 12 h was selected to further study the effects of different concentrations of SMX and EHMC on the nitrate reduction process.

As shown in Figure 3, the single and combined pollutants inhibited the dissimilatory nitrate reduction processes of the sediments from the Yarlung Zangbo River. The denitrification rates were significantly inhibited when the concentration of SMX were higher than, or equal to 1 μg L^−^^1^. The anammox rates were obviously decreased when the concentration of SMX was higher than 0.1 μg L^−^^1^. In addition, both rates were decreased corresponding to increases in the SMX concentration, and the concentration dependence was obvious. The results showed that the higher concentration of SMX, the stronger inhibition of the dissimilatory nitrate reduction processes. The maximum denitrification inhibition rate was up to 95.8%, which means that an antibiotic input will effectively reduce nitrogen removal in the aquatic environment. Antibiotics can cause phylogenetic structure alterations and ecological function disturbances in micro-ecosystems [41,42]. Antibiotics can also promote the development of resistance genes in the aquatic environment [43,44]. These results highlight the risk of antibiotic residue due to nitrogen removal in the high-altitude aquatic environment. It has been reported that the sulfamethazine added to the sediments, collected from the Yangtze River, stimulated the denitrifying bacteria to develop antibiotics resistance and antibiotic resistance genes [19]. This phenomenon did not appear in our results. According to the existing data, the pollution from antibiotics and nitrogenous compounds is more serious in the Yangtze River than in the Yarlung Zangbo River [18,45]. In addition, the different of microbial distributions of the Yarlung Zangbo River and the Yangtze River may account for the absence of antibiotic resistance genes in this study [15,46].

EHMC significantly decreased the anammox rates at all the test concentrations (Figure 3), and the suppression rates ranged from 59% to 71%, but no obvious concentration dependence was found. However, denitrification rates were decreased by EHMC treatment at two lower concentrations (0.01 and 0.1 μg L^−^^1^) and the highest concentration (100 μg L^−^^1^). The maximum inhibition rate was up to 90%. The denitrification rate had significant concentration differences in the EHMC group. At low concentrations (0.01 μg L^−^^1^), EHMC biodegradation consumed nutrients in water, thus inhibiting denitrification. While in the medium concentrations, the carbon and nitrogen source were supplemented by the degradation of EHMC. As the concentration continues to rise to 100 μg L^−^^1^, denitrification was suppressed again. The inhibition of EHMC on microbial gene expression could explain the decrease of denitrification rates [29]. The denitrification rates were significantly decreased by the SMX and EHMC treatments, and the inhibition rates ranged from 42% to 86%. Compared with the single pollutant exposure, the mixtures induced stronger inhibition effects in most cases. One possible reason is that SMX and EHMC influence the denitrification process in different stages, which leads to a superposition of the inhibition effects. Regarding the anammox rates, the combined treatments caused an inhibitory effect that was similar to that of EHMC alone, and the inhibition rates ranged from 50% to 65%. At present, the effects of personal care products on nitrogen transformation have not been reported, and it is noteworthy that our results, for the first time, demonstrate the important impact of the OUVFs on the nitrate reduction processes, especially the superposition effects caused by OUVFs combined with antibiotics. Evaluating the risks of mixtures is of great importance because pollutants usually coexist in the real aquatic environment rather than occurring as single entities.

### 3.3. Influence of SMX and EHMC on N_2_O Release 

N_2_O, referred to as a greenhouse gas, is an intermediate of the denitrification processes. Compared to the control group, the production of N_2_O was significantly increased in all groups in the course of the incubation (Figure 4). The N_2_O release rates increased with the SMX concentrations and reached twice that of the control at the highest SMX concentrations. In the EHMC group, the N_2_O release rates were 2.5 times that of the control at 0.01 and 10 μg L^−^^1^, and no significant difference was found between these tests and the tests with the highest concentration. However, the release rates of N_2_O produced by EHMC (0.1 and 1 μg L^−^^1^) were markedly decreased when compared with those produced by the other concentrations. In the combined pollutants groups, the releases of N_2_O increased with the increase in the exposure concentration, revealing concentration-dependent trends. The N_2_O release rates increased by 1.6 to 2.7 times that of the control, which were higher than that of SMX alone and close to that of EHMC alone. The release rates of N_2_O in the denitrification process are determined by a balance between nitrite reduction to N_2_O and N_2_O reduction to N_2_ [6]. The N_2_O accumulation that was induced by SMX and EHMC (alone or in a combination) indicated that the two PPCPs may either inhibit N_2_O reduction to N_2_ or stimulate N_2_O production. It has been reported that some antibiotics (tetracycline, sulfamethazine and chloramphenicol) increased the release of N_2_O by promoting soil respiration or accumulation of nitrite [19,23,47], while sulfamethoxazole and chloramphenicol were found to reduce N_2_O release due to inhibition of denitrifying enzyme activities [48]. In the current study, EHMC and the mixtures were noticed to produce an inductive effect on N_2_O release. The continuous input of PPCPs in the high-altitude rivers could induce more N_2_O release, and this might increase the pressure of global warming.

### 3.4. The Influence of SMX and EHMC on the Abundance of NirS and NosZ

The changes in the abundance values of the *nirS* and *nosZ* in the treatments with SMX and EHMC (alone or in combination) are shown in Figure 5. In comparison with the control group, the abundance values of both genes were decreased significantly by all treatments. The repression of the *nirS* and *nosZ* gene expression values corresponded to increases in the SMX and EHMC concentrations, with the exception of one or two of the lowest concentrations. The maximum inhibition rates were 75% for *nirS* and 78.0% for *nosZ*. In the SMX group, there was no significant difference in *nirS* abundance when the concentration was less than 1 μg L^−^^1^. The abundance of *nirS* remained stable when the SMX concentration was greater than or equal to 10 μg L^−^^1^. In the EHMC group, low concentration (0.01 μg L^−^^1^) and high concentration (100 μg L^−^^1^) showed stronger inhibition on abundance of *nirS* and *nosZ*. The Pearson correlation coefficient indicates that the release of N_2_O is significantly correlated with *nirS* and *nosZ* (Table 3). The inhibition of *nirS* and *nosZ* by low and high concentration of EHMC could explain the N_2_O concentration difference in the EHMC group. In the SMX and EHMC group, it basically conformed to the rule that the gene abundance inhibition rate increased with the increase of pollutant concentration. In general, the inhibition effects of the mixtures on *nirS* were stronger than those of the single pollutants exposures. The *nirS* genes, which encode nitrite reductase, are closely related to the nitrite reduction to nitric oxide step. Additionally, the *nosZ* genes encode nitrous oxide reductase, which is directly involved in reducing nitrous oxide to nitrogen gas. The inhibitory rates of EHMC on *nosZ* were larger than its inhibition of *nirS*, which may explain why N_2_O accumulated in the incubation experiments with the SMX and EHMC treatments. The *nosZ* genes are related to the reduction of N_2_O to N_2_, and the *nirS* genes are associated with N_2_O production. The inhibition effect of two PPCPs on *nosZ* abundance was more effective than that of *nirS*. This result indicated that the two PPCPs may inhibit the reduction of N_2_O to N_2_ more than the generation of N_2_O. Moreover, the inhibition of the *nirS* and *nosZ* genes induced by the two PPCPs can explain the decreases in the denitrification rates described above. Similar results were reported by Hou et al. [19,49], where the concentration of antibiotics were related to the decrease of the nitrogen transform rates. In addition, the synthesis of folic acid necessary for bacterial growth is affected by SMX [19]. This may be a reason that the denitrifying gene expression decreased remarkably during the incubation process. Our results showed that EHMC as well as SMX decreased the abundance of *nirS* and *nosZ* genes and thus inhibited dissimilatory nitrate reduction and increased N_2_O production in the high-altitude rivers. In this study, the combinations of SMX and EHMC produced greater inhibitory effects on the *nirS* gene expression and produced greater inhibitory effects on the denitrification rates than the individual exposures. The joint effects of the coexistent PPCPs on the nitrogen transformation processes in the natural aquatic environment require further study.

## 4. Conclusions

This study demonstrates that the denitrification rates and associated N_2_O release rates are intimately related to the presence of PPCPs in a high-altitude river. The rates of denitrification and anammox were inhibited by the sulfamethoxazole (SMX) treatments (1–100 μg L^−^^1^), which may be due to the inhibitory effect of this antibiotic on the nitrate reducing microbes. The inhibitory effect of 2-ethylhexyl-4-methoxycinnamate (EHMC) on the anammox process was even stronger than its effect on the denitrification process. SMX and EHMC in combination showed a superposition effect on the denitrification processes. In addition, SMX alone and EHMC alone and in combination will decrease the expression levels of the denitrifying functional genes and thus promote the release of N_2_O. Stimulating production of N_2_O from exposure to emerging pollutants may increase the contribution of high-altitude rivers to global warming. In summary, the results of this study provide data that demonstrates the effects of PPCPs on nitrogen transformation and highlights the environmental risks due to the presence of emerging pollutants in high-altitude rivers.

## Figures and Tables

**Figure 1 ijerph-17-01822-f001:**
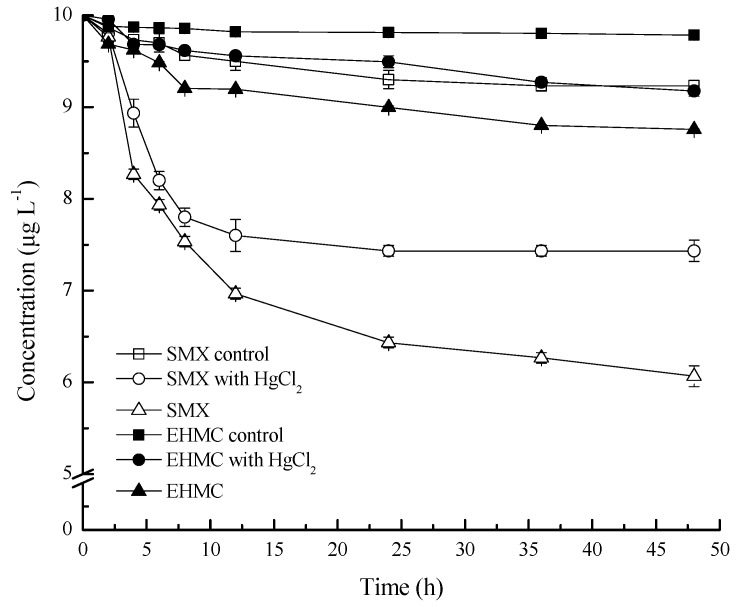
Temporal changes in the SMX and EHMC concentrations in water. The control group was no sediments; abiotic was the sediment containing HgCl_2_; biotic was the sediment with untreated. The values are the means and the error bars represent the standard deviations (n = 3).

**Figure 2 ijerph-17-01822-f002:**
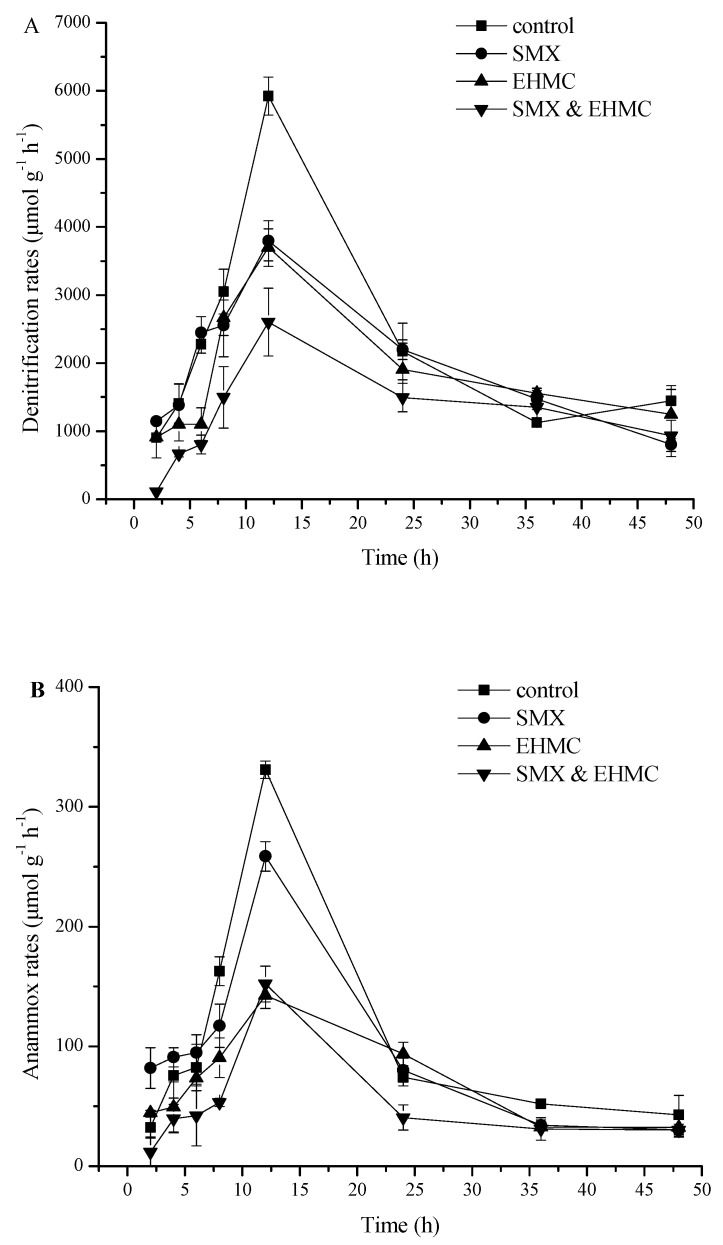
Time series changes in the rate of denitrification (**A**) and anammox (**B**) under the effect of PPCPs. Control indicates the group with no injected pollutants; SMX, EHMC, and SMX&EHMC treatments mean that a group was spiked with SMX alone, EHMC alone or a combination of SMX and EHMC (final concentration of 1 µg L^−^^1^). The values are the means and the error bars represent the standard deviations (n = 3).

**Figure 3 ijerph-17-01822-f003:**
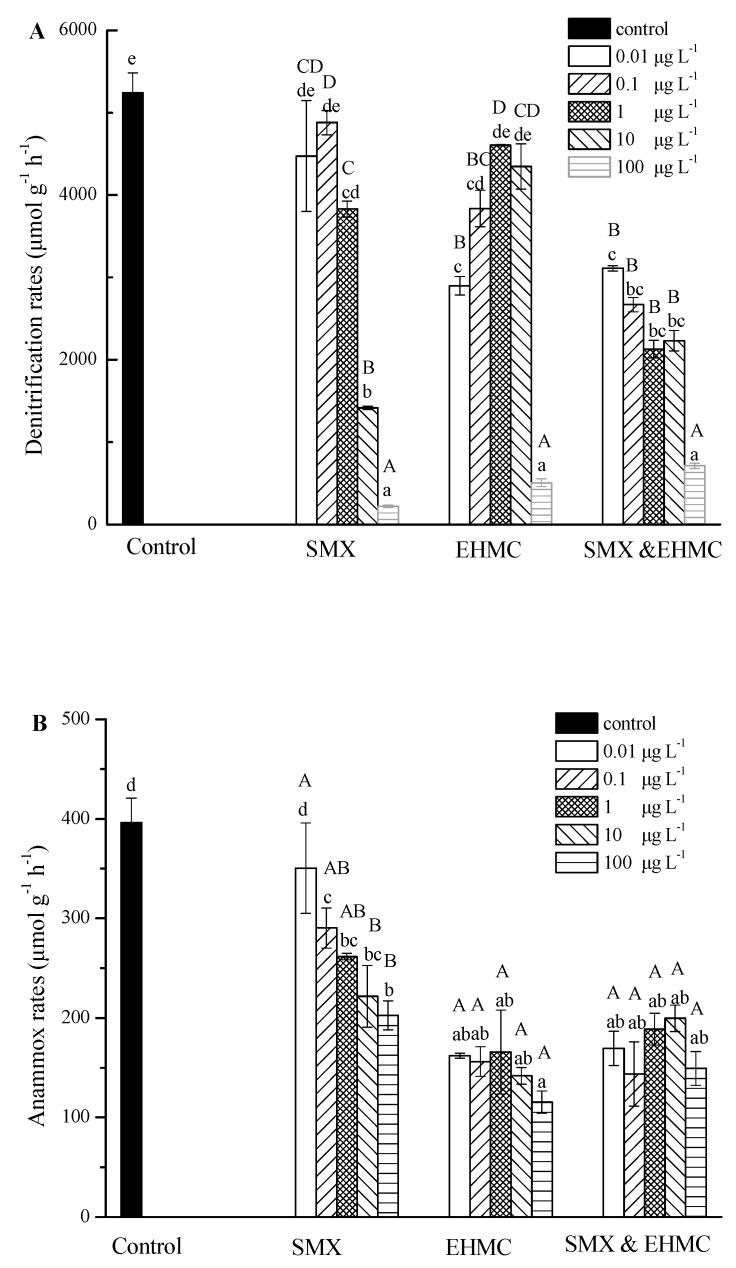
The concentration effects of SMX and EHMC (alone or in a combination) on denitrification (**A**) and anammox (**B**) rates. The values are the means and the error bars represent the standard deviations (n = 3). The treatment groups sharing different lowercase letters represent significant differences (*p* < 0.05). The uppercase letters indicate significant differences between concentrations in the same treatment (*p* < 0.05).

**Figure 4 ijerph-17-01822-f004:**
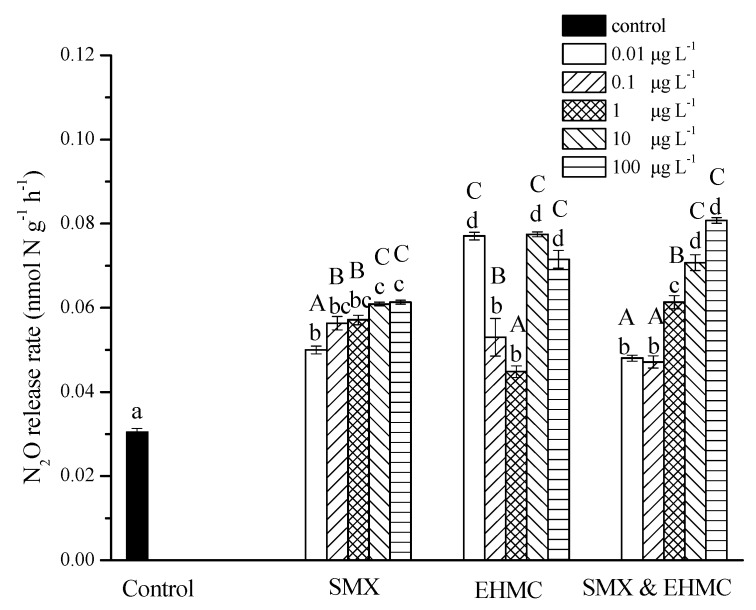
The effects of SMX and EHMC on N_2_O release. The values are the means and the error bars represent the standard deviations (n = 3). The treatment groups sharing different lowercase letters represent significant differences (*p* < 0.05). The uppercase letters indicate significant differences between concentrations in the same treatment (*p* < 0.05).

**Figure 5 ijerph-17-01822-f005:**
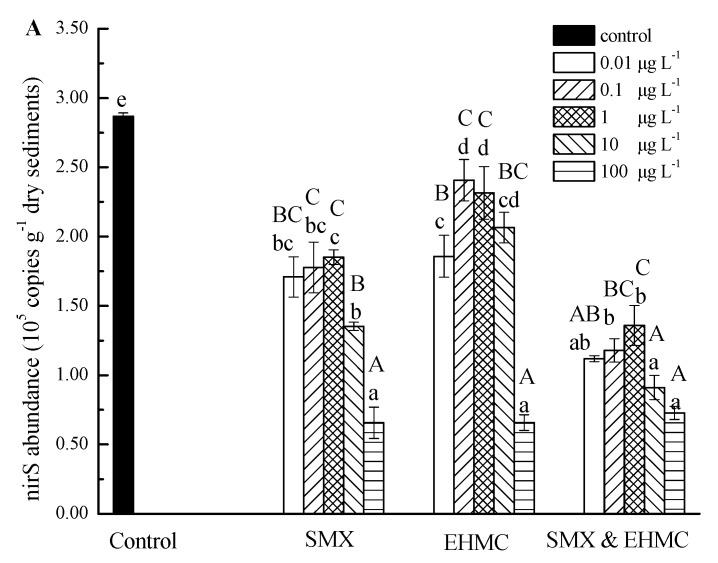
The effects of SMX and EHMC on *nirS* (**A**) and *nosZ* (**B**) genes. The values are the means and the error bars represent the standard deviations (n = 3). The treatment groups sharing different lowercase letters represent significant differences (*p* < 0.05). The uppercase letters indicate significant differences between concentrations in the same treatment (*p* < 0.05).

**Table 1 ijerph-17-01822-t001:** The water quality parameters and sediment physicochemical properties.

Parameter in Water	Value	Parameter in Sediment	Value
Temperature (°C)	13.7	TN (mg kg^−1^)	354.87
NH_4_^+^ (mg L^−1^)	0.34	NH_4_^+^ (mg kg^−1^)	4.69
NO_3_^−^ (mg L^−1^)	0.45	NO_3_^−^ (mg kg^−1^)	2.37
NO_2_^−^ (mg L^−1^)	0.07	NO_2_^−^ (mg kg^−1^)	0.21
SMX (ng L^−1^)	2.3	SMX (ng g^−1^)	4.36
EHMC (ng L^−1^)	0.9	EHMC (ng g^−1^)	16.4

**Table 2 ijerph-17-01822-t002:** Primers, thermal profiles and parameters for qRT-PCR quantification of *nirS* and *nosZ*.

Target	Primer	Nucleotide Sequence	Thermal Profile	Reference
*nirS*	cd3aF	GTSAACGTSAAGGARACSGG	95 °C, 3 min, 1cycle, 95 °C for 30 s, 56 °C for 30 s, 72 °C for 30 s, 40 cycles	[36]
	R3cd	GASTTCGGRTGSGTCTTGA		
*nosZ*	nosZ2F	CGCRACGGCAASAAGGTSMSSGT	95 °C, 3 min, 1 cycle, 95 °C for 30 s, 56 °C for 30 s, 70 °C for 30 s, 40 cycles	[37]
	nosZ2R	CAKRTGCAKSGCRTGGCAGAA		

**Table 3 ijerph-17-01822-t003:** Pearson correlation coefficients between the expression of genes *nirS*, *nosZ*, denitrification, anammox, and N_2_O release.

Varibles	Denitrification	Anammox	*nirS*	*nosZ*	N_2_O Release
Denitrification	1	0.403 ^**^	0.684 ^**^	0.622 ^**^	−0.409 ^**^
Anammox		1	0.437 ^**^	0.570 ^**^	−0.519 ^**^
*nirS*			1	0.789 ^**^	−0.514 ^**^
*nosZ*				1	−0.657 ^**^
N_2_O release					1

^**^. Correlation is significant at the 0.01 level (2-tailed).

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
