# Peer review of "Effects of Sulfamethoxazole and 2-Ethylhexyl-4-Methoxycinnamate on the Dissimilatory Nitrate Reduction Processes and N_2_O Release in Sediments in the Yarlung Zangbo River"

_ijerph, 2020, doi:10.3390/ijerph17061822_

Round 1

Reviewer 1 Report

Major issues that should be addressed for revising the manuscript

  1. Why is nitromonas (nitrobacter) not discussed in this manuscript?
  2. Why did the authorschoose the month of May to collect near-bottom water and surface sediments from the Linzhi Section of the Yarlung Zangbo River?
  3. The authors should provide the calculation method of removal rate and adsorption rateof sulfamethoxazole (SMX) and 2 - ethylhexyl - 4 - methoxycinnamate (EHMC).
  4. The language of the manuscript should be carefully polished.

Minor issues that should be addressed

  1. Figures 1 and 2: "times" should be "time".
  2. Please note the upper case and lower casein the keywords.
  3. Page 1, lines 35 and 38: please correct the language.
  4. Page 2, line 65: please check “the relevant microbes has not been reported”.
  5. Page 3, line 114: please check “centrifuged at 2000 ×g (10 min) to yield the sediments and supernatants”.
  6. Page 4, line 156: please check Eq. (3).
  7. Page 4, line 164: please correct the language.
  8. Page 5, line 175: please check Table 2,such as “10 min at 95°C, 40 cycles of 30 s at 95°C”.
  9. Page 5, line 188: please check “the concentrations of EHMC”.
  10. Page 6, line 198: please check “The decrease in SMX in the control group (<8%)”.
  11. Page 6, line 228: please check “SMX concentrations ≥0.1 μg L-1”.
  12. Page 7, line 264: please check “all the groups in the in the course of the incubation”.
  13. Page 9, line 305: please check “In addition, In addition,”.
  14. Page 9, line 320: please correct the language.
  15. Please carefully check and correct the references.

Reviewer 2 Report

The authors have assessed the influence of SMX and EHMC on the dissimilatory nitrate reduction and N2O release in sediments of the Yarlung Zangbo river. Overall, this well-written study is very interesting and reports novel findings of great significance. Results are clearly presented and discussed and the experimental design was adequately conceived. Still, before proceeding with its publication, I'd like the authors to do the following improvements:

  • While evaluating the effects on SMX and EHMC on different parameters (denitrification; anammox; N2O release; and genes expression), Figures 3-5: Instead of limiting the statistical analysis to a global comparison across treatments (control, SMX, EHMC, and SMX+EHMC); the authors should also carry out additional ANOVA analyses to determine significant differences between concentrations (0.01, 0.1, 1, 10 and 100 μg L-1) for each treatment - you can use a system of uppercase letters to indicate these differences. Given that each treatment represents a different population with different variances, the differences found won't always match the lowercase letters obtained for the global ANOVA. Thereby, you should incorporate these new results to your discussion.
  • The paper, especially the discussion, would benefit from the addition of a Pearson correlation table establishing the relationship between the expression of genes nirS, nosZ, denitrification, anammox, and N2O release.
  • Page 8, Line 269: "...the release rates of N2O produced by 0.1 and 1 μg L-1 of EHMC were markedly decreased when compared
    270 with those produced by the other concentrations" - true, but the authors need to elaborate on this. Please explain why (increase gene expression at those concentrations? perhaps the pearson correlation table could help?)

Reviewer 3 Report

Reviewers' comments:

This paper which was titled “Effects of Sulfamethoxazole and 2-Ethylhexyl-4-Methoxycinnamate on the Dissimilatory Nitrate Reduction Processes and N2O Release in Sediments in the Yarlung Zangbo River” describes the study of PPCPs on the nitrogen transformation in high-altitude river sediments. In this study, sediment slurry experiments combined with 15N tracer techniques were conducted. The influence of two typical PPCPs (Sulfamethoxazole and 2-Ethylhexyl-4-Methoxycinnamate) on denitrification and the anaerobic ammonium oxidation (anammox) process and the resulting N2O release in the sediments of the Yarlung Zangbo River were investigated. The results show that both SMX and EHMC can inhibit denitrification and anammox process, reduce the expression level of denitrification functional genes and promote N2O release. The data provided in this study demonstrates the impact of PPCPs on nitrogen conversion and highlights environmental risks caused by emerging pollutants in high-altitude rivers. We believe that this manuscript has certain innovation and a strong practical guiding significance for the development of Yarlung Zangbo River but still has some shortcomings. I recommend this manuscript to be published in this journal after some revision. My detailed comments are as follows:

  1. The language of the article should be strengthened. For example, line 41, change “at the present time” to “at present”, line 71, delete “individual”, the above are tautology. Line 188, add “the/an” in front of “abiotic group”. Check the manuscript with native English speaker.
  2. References need to be updated, especially in the introduction, and only half of the references have been in the last 5 years.
  3. In the introduction section, the author describes the research and classification of PPCPs as emerging pollutants but did not point out why sulfamethoxazole and 2-Ethylhexyl-4-Methoxycinnamate should select as the research object. Throughout the manuscript, PPCPs have used as the research object many times. There may be general problems and misunderstandings to readers. Therefore, it is recommended that the research object and research purpose of the article be unified and accurate.
  4. Table 2 should further standardize and align.
  5. In section 3.1, the authors designed experiments to study the effects of microorganisms on the degradation of sulfamethoxazole and 2-Ethylhexyl-4-Methoxycinnamate. What is the relation to the experiments in the following sections? Please explain the results and discussion.

Round 2

Reviewer 1 Report

Minor issues that should be addressed for revising the manuscript.

  1. The language of this manuscript still needs to be improved.
  2. Page 3, line 126: “2000 ×g (10 min)”should be “2000 × g (10 min)”.
  3. Please check the font on lines 140-143.
  4. Page 4, line 140: “Where”should be “where”.
  5. Page 4, line 143: please check “4. Slurry incubation experiments for the nitrate reduction processes”.
  6. Page 4, ling 154: please check “withoutpollutants”.
  7. Page 5, line 179-180: please check the punctuation afterEq (8).
  8. Page 5, line 186: please check “ofSMX”.
  9. Please check the unit of temperature in table 2.
  10. Page 8, line 278: please check font “to 100 μg L-1”.
  11. Page 9, line 331: “(0.01μg L-1)”should be “(0.01 μg L-1)”.

Reviewer 2 Report

The authors have significantly improved the manuscript. I have only one last concern regarding the Pearson correlation analysis, which must be addressed - more specifically:

- The authors forgot to add the information about the Pearson correlation analysis in the "Statistical Analysis" section.

- Table 3: The Pearson correlation values lack the indicatives of significant correlations (0.05 or 0.01 level), where applicable. See the following example: https://spss-tutorials.com/img/APA-format-correlation-table.png

Reviewer 3 Report

I have gone through the revised manuscript. I believe that the authors have provided sufficient explanation for the questioned raised by the reviewers. As such, I am convinced that the data presented in the manuscript are robust and the conclusions are justified. I recommend the publication of this manuscript in this jounral.

Author Response

Thank you for your valuable comments on our manuscript. We revised some sentences and formatting errors in the manuscript. We sincerely hope that this manuscript will be accepted after revision.